# Tumor Necrosis Factor-Alpha Inhibitor Use and Malignancy Risk: A Systematic Review and Patient Level Meta-Analysis

**DOI:** 10.3390/cancers17030390

**Published:** 2025-01-24

**Authors:** Conor B. Driscoll, Jordan M. Rich, Dylan Isaacson, Joseph Nicolas, Yu Jiang, Xinlei Mi, Christopher Yang, Victoria Kocsuta, Regine Goh, Niti Patel, Eric Li, Mohammad Rashid Siddiqui, Travis Meyers, John S. Witte, Linda Kachuri, Hui Zhang, Molly Beestrum, Philip Silberman, Edward M. Schaeffer, Shilajit D. Kundu

**Affiliations:** 1Department of Urology, Northwestern University Feinberg School of Medicine, 676 North St Clair Street, Suite 2300, Chicago, IL 60611, USA; jorrich@umich.edu (J.M.R.); joseph.nicolas@northwestern.edu (J.N.); christopher.yang@northwestern.edu (C.Y.); victoria.kocsuta@northwestern.edu (V.K.); gohregine@gmail.com (R.G.); eric.li1@northwestern.edu (E.L.); sidr87@gmail.com (M.R.S.); e-schaeffer@northwestern.edu (E.M.S.); shilajit.kundu@nm.org (S.D.K.); 2Department of Epidemiology and Population Health, Stanford University, 300 Pasteur Drive, Stanford, CA 94305, USAjswitte@stanford.edu (J.S.W.); lkachuri@stanford.edu (L.K.); 3Division of Biostatistics, Department of Preventative Medicine, Northwestern University Feinberg School of Medicine, 680 N. Lake Shore Dr, Suite 1400, Chicago, IL 60611, USA; xinlei.mi@northwestern.edu (X.M.); hzhang@northwestern.edu (H.Z.); 4Department of Medicine, Northwestern University Feinberg School of Medicine, 676 North St. Clair Street, Arkes Suite 2330, Chicago, IL 60611, USA; niti.patel@nm.org; 5Department of Epidemiology and Biostatistics, University of California at San Francisco, 550 16th St., Floor 2, San Francisco, CA 94143, USA; travis.meyers@ucsf.edu; 6Department of Biomedical Data Science, Stanford University, 1265 Welch Road MC5464MSOB West Wing, Third Floor, Stanford, CA 94305, USA; 7Galter Health Sciences Library and Learning Center, Northwestern University, 303 E Chicago Ave #2-212, Chicago, IL 60611, USA; molly.beestrum@northwestern.edu; 8Department of Information Technology, Northwestern University Feinberg School of Medicine, 710 N. Lake Shore Dr, Abbott Hall, 4th Floor, Chicago, IL 60611, USA; psilber1@nm.org

**Keywords:** meta-analysis, cancer, TNF-alpha, adalimumab, infliximab, etanercept, certolizumab, golimumab

## Abstract

TNF-α inhibitors are widely prescribed immunosuppressive medications for multiple chronic inflammatory disorders. Chronic immunosuppression can lead to malignancy due to decreased immune surveillance, but TNF-α inhibitors have been found to be largely safe aside from the development of non-melanoma skin cancer. However, these studies all have short-term follow-up, rely on spontaneous reporting systems, and investigate a disease or drug subset. Previous meta-analyses similarly have investigated either interventional or observational studies only. Here, we reviewed over 17,000 abstracts with the most stringent inclusion criteria in the published literature to analyze 46 interventional studies and 10 observational studies for malignancy risk in patients exposed to TNF-α inhibitors. Using patient-level data, our findings for both interventional and observational studies found no statistically significant increased risk of malignancy but with poor quality of adverse event reporting. This is the most comprehensive and up-to-date meta-analysis and underscores the need for high-quality studies with a longer follow-up of these patients.

## 1. Introduction

Over the last two decades, tumor necrosis factor-alpha inhibitors (TNF-Is) have become standard therapies for chronic inflammatory disorders, with an ongoing expansion of indications and off-label applications [1]. Presently, five TNF-Is have been approved by the U.S. Food and Drug Administration (FDA) for the treatment of inflammatory conditions: Adalimumab (ADA) and golimumab (GLM) are fully humanized monoclonal tumor necrosis factor-alpha (TNF-α) IgG1 antibodies, infliximab (INX) is a mouse/human chimeric monoclonal TNF-α IgG1, certolizumab pegol (CTZ) is a PEGylated Fab’ fragment of a TNF-α monoclonal antibody, and etanercept (ETN) is a fusion protein of the human tumor necrosis factor receptor II extracellular domain and the IgG1 Fc region [2]. The wide-ranging and effective use of these medications is typified by the status of adalimumab—the most profitable drug worldwide for much of the past decade [3,4,5]. Though their structures and exact functions vary [6], these agents all inhibit the binding of soluble TNF-α to its receptor [2], inhibiting the downstream production of pro-inflammatory cytokines and chemokines, the upregulation of adhesion molecules, and the activation of peripheral blood mononuclear cells [1]. Concurrent with the roles of chronic inflammation in promoting carcinogenesis [7] and of immune activation in clearing malignant cells, it remains unclear whether TNF-α exerts tumorigenic or anti-neoplastic activity, or both depending on the host immunologic context [8,9,10].

Multiple groups have evaluated the relationship between TNF-I exposure and malignancies. The strongest data are those associating TNF-I use with non-melanoma skin cancers (NMSCs), for which institutional cohort studies [11], registry-based studies [12,13,14], and meta-analyses [15,16,17] have demonstrated an increased risk. Some reports have suggested a higher incidence of lymphomas in patients taking TNF-I [12,18,19,20,21]; however, these analyses were potentially confounded by associations with underlying disease states and other medication use [10,22], and their findings have largely been outweighed by several studies in diverse populations demonstrating no increased risk [23,24,25,26,27,28,29]. For patients with pre-existing malignancies, current clinical practice states that biologic therapy, such as that with TNF-I, is not absolutely contraindicated and must be approached on a case-by-case basis with the patient [30].

TNF-I use is frequently presumed to have no influence on the development of other malignancies, as demonstrated by the results of multiple clinical trials, registry-based studies, and meta-analyses conducted over the last decade [17,27,31,32,33,34,35]. However, these studies have key limitations—most clinical trials of TNF-I were primarily designed to assess drug efficacy, and many co-administered medications such as methotrexate and were not powered specifically to detect incident malignancies [36]. Studies drawing from administrative databases rely on spontaneous adverse event reporting and algorithms for identifying malignancies, which risks underestimating overall cancer incidence, false positives, and bias from the systematic misclassification of suspected cases [37]. Existing meta-analyses incorporate these limitations of primary studies and have been constrained in evaluating only specific subsets of approved TNF-I [17,38], single malignancy types [16], second malignancies [35] or specific inflammatory disease states [15,16,31,32,33,34,35,38]. These meta-analyses have been further restricted to include only investigator-controlled interventional studies [17,31,32,33,34] or observational registry-based analyses [27]. In parallel, data from the FDA Adverse Events Reporting System describe cancers associated with TNF-I use in excess of the subset reported in the scientific literature [39], suggesting a potentially unappreciated relationship.

Given the limitations of current meta-analyses, we sought to comprehensively evaluate associations between all approved TNF-Is and the development of malignancies. In this work, we systematically review contemporary interventional and observational studies and perform meta-analyses on these two groups of studies. To the extent of our knowledge, this is the broadest attempt to evaluate the relationship between TNF-I and incident malignancies to date.

## 2. Materials and Methods

### 2.1. Relevant Publication Search

We searched MEDLINE for studies published between January 1996 and January 2020 evaluating approved TNF-I using the medical subject headings (MeSH) “TNF-alpha Inhibitors”, “adalimumab”, “certolizumab”, “etanercept”, “golimumab”, “infliximab”, “safety”, “side effects”, “toxicity”, “neoplasms”, “cancer”, “malignan*”, “adverse”, “longitudinal studies”, “short term”, and “long term”. We also searched using keywords for TNF-I trade names. We searched ClinicalTrials.gov using the keywords “TNF-alpha Inhibitors” and each of the five individual drug names. We replicated this search process in EMBASE (Elsevier), the CENTRAL Register of Controlled Trials (Wiley), the Cochrane Library (Wiley), Scopus (Elsevier), and the gray literature sources. The full search strategy, using infliximab as an example, is reproduced in the Supplementary Methods.

### 2.2. Inclusion and Exclusion Criteria

Criteria for inclusion in the meta-analysis were as follows: (1) Clinical trials or population-based observational studies of TNF-I approved for use in the United States, Europe, or Asia. (2) Presence of a safety analysis delineated by individual TNF-I. (3) Use of a TNF-I in a systemically absorbed formulation, such as intravenous or subcutaneous. The two medications administered intravenously are INX (3 mg/kg every 8 weeks) and GLM (100 mg every 4 weeks after induction) are administered intravenously. The three medications administered subcutaneously are ADA (40 mg every other week), ETN (total of 50 mg weekly), and CTZ (total of 400 mg every 4 weeks after induction).

Exclusion criteria were as follows: (1) Non-experimental study design—practice updates, expert opinions, letters, abstracts, case series, or case reports. (2) Publication in a language other than English. (3) Studies conducted on non-human subjects. (4) Analyses focusing on a single non-malignancy adverse event (i.e., tuberculosis). (5) Studies not performed using primary data (i.e., insurance claims databases, meta-analyses). (6) Single-dose studies assessing immediate infusion side effects. (7) Trials in which experimental medications were co-administered with TNF-I. (8) TNF-I withdrawal studies. (9) Studies incorporating a crossover component between a TNF-I and another experimental drug.

### 2.3. Data Extraction

Candidate studies were classified as interventional if prospectively assessing the effects of a TNF-I prescribed per experimental design, or observational if investigators played no role in dictating TNF-I administration. Studies were evaluated for TNF-I type, dosage and disease indication, study design, inclusion/exclusion criteria, sample size and duration of follow-up, patient demographics, incident malignancy cases, other adverse events, subject withdrawal rate, reported conflicts of interest, and funding sources. Two observers (C.D. plus R.G., N.P., D.I., E.L., or M.R.S.) independently extracted the data; differences were resolved by consensus.

### 2.4. Quality of Adverse Event and Risk of Bias Classification

We classified the adequacy of blinding as present if explicitly reported, “unknown” if not reported, or “not applicable” for single-arm studies. We scored drug safety reporting as “adequate”, “partially adequate”, or “inadequate” as described by Ioannidis and Lau [40]. Reports explicitly noting no adverse events were scored as “none”. Studies were classified as having “adequate” quality of adverse event reporting if they graded severity using a graduated toxicity scale and reported adverse event incidence in each study arm versus “partially adequate” if they failed to distinguish between moderate and severe toxicities or otherwise did not meet criteria for the “adequate” or “inadequate” categories. Studies were classified as “inadequate” if they did not detail the specific classes of adverse events, provided only generic statements, or omitted comment entirely. The quality of adverse event classifications for studies included in the meta-analyses are listed in Appendix A.

Risk of bias was assessed with the modified Cochrane analysis tool [41] for randomized studies and the ROBINS-I analysis tool for non-randomized studies [42]. Risk of bias assessments for included studies are detailed in Appendix A.

### 2.5. Statistical Analysis

For studies meeting criteria, we estimated malignancy rate ratios (MRRs) for the TNF-I-exposed vs. control groups with direct calculation. For studies evaluating multiple TNF-Is, each individual TNF-I-exposed group was analyzed separately against the control group. MRRs were estimated for overall (aggregate) malignancies excluding NMSC and for individual cancer subtypes [43]. In studies reporting no malignancies in either the TNF-I-exposed or control group, a continuity correction of 0.5 was applied to both groups. Studies reporting no malignancies in both groups were excluded from analysis. We used a random-effects meta-analysis model to account for potential heterogeneity between studies and to estimate pooled effects (95% confidence intervals) for rate ratios. The overall effect estimates are presented as a weighted mean of the study-specific estimates [44]. *p* values < 0.05 were considered statistically significant throughout the study and determined in a two-sided manner. Some studies involved multiple drugs and utilized shared unexposed controls, and there can be overlap between the contributed patient-years across multiple drug arms, from which dependencies can arise. In the cross-drug meta-analysis of the observational studies, we took such dependencies into account and used the rma.mv function in the metaphor package in R to model the non-independence structure induced by overlapping unexposed controls in the same study [45].

Forest plots were created using a standard rate scale to assess for publication bias. Studies with no incident malignancies other than NMSC were excluded from these plots. Study heterogeneity was evaluated using Cochran’s Q statistic and the I2 statistic. I2>50% was taken to represent moderate heterogeneity and I2>75% to represent high heterogeneity.

Publication bias within studies was assessed with a funnel plot using a random-effects model. Rank correlation tests were used to assess for funnel plot asymmetry for the observational and interventional studies of each TNF-I and for observational and interventional studies aggregating all TNF-Is. Statistical analyses were carried out using R 4.0.2 (R Foundation, Vienna, Austria) and the metafor package [46].

## 3. Results

### 3.1. Search Results

We identified 16,150 candidate publications, yielding 55 unique studies for quantitative analysis (Figure 1). In total, 45/55 (82%) of the identified studies were interventional [47,48,49,50,51,52,53,54,55,56,57,58,59,60,61,62,63,64,65,66,67,68,69,70,71,72,73,74,75,76,77,78,79,80,81,82,83,84,85,86,87,88,89,90,91], and 10/55 (18%) were observational [25,92,93,94,95,96,97,98,99,100]. The PRISMA 2020 flowchart is included as Figure 1.

### 3.2. Interventional Study Results

The 45 interventional studies incorporated into the meta-analysis contributed a total of 22,652 patients with a median exposure of 41 weeks (IQR of 24–60 weeks) (Table 1). In total, seventeen of these studies reported on ADA (one study contained two separate randomized trials in the manuscript) [61], twelve on INX, four on ETN, six on CTZ, and seven on GLM. The median number of subjects exposed to a TNF-I was 360 (IQR of 216–634 patients). The median length of follow-up was 41 weeks (IQR of 24–64 weeks). These studies contributed a total of 146 malignancies over 15,120 patient-years of exposure to TNF-I: thirteen lymphomas, twelve breast cancers, twelve lung cancers, nine melanomas, eight colorectal cancers, five pancreatic cancers, four head and neck cancers, four ovarian cancers, three gastric cancers, three cervical cancers, two brain cancers, two renal cell carcinomas, two prostate cancers, two unspecified adenocarcinomas, one cholangiocarcinoma, one leukemia, one gastrointestinal carcinoid tumor, one bladder cancer, one testicular cancer, one endometrial cancer, and fifty-nine unspecified cancers (Appendix A). In total, 62 NMSCs were also recorded but not used in the meta-analysis.

There were 29 malignancies recorded over 4695 patient-years of placebo and/or non-TNF-I drug exposure: four breast cancers, three lymphomas, three prostate cancers, two cervical cancers, two lung cancers, one colorectal cancer, one esophageal cancer, one thyroid cancer, and twelve unspecified cancers (Appendix A). Nine NMSC were also recorded but not used in the meta-analysis.

The malignancy rate ratio for any malignancy was MRR 1.04 (95% CI 0.71–1.51) (Figure 2). For individual drugs, malignancy rate ratios were as follows: ADA (MRR 1.19, 95% CI 0.44–1.03), INX (MRR 1.35, 95% CI 0.67–2.72), ETN (MRR 0.80, 95% CI 0.22–2.88), CTZ (MRR 1.60, 95% CI 0.72–3.57), and GLM (MRR 1.49, 95% CI 0.59–3.77) (Figure 3A–E). Funnel plots did not demonstrate statistically significant asymmetry for overall TNF-I exposure or for any individual TNF-I exposure (Appendix A).

In the interventional studies, only 9/45 (20%) had adequate quality of adverse event reporting according to the Ioannidis and Lau criteria, and 44/45 (98%) had either industry or unknown funding sources (Table 2). In total, 38/45 (84%) of the interventional studies had a low risk of bias due to high-quality randomization per the Cochrane criteria (Appendix A).

### 3.3. Observational Study Results

The ten observational studies meeting the criteria for the meta-analysis contributed 34,866 patients with a median of 83 weeks of exposure (IQR 52–171 weeks) (Table 1). Five reported on ADA, five on INX, and five on ETN, with four of those studies reporting on more than one TNF-I. For these studies, the median number of patients exposed to a TNF-I was 1606 patients (IQR 433–3057 patients). The median length of follow-up was 82 weeks (IQR 54–171 weeks). There were 108 malignancies in 28,796 patient-years of TNF-I exposure: thirty-seven lymphomas, ten lung cancers, three head and neck cancers, three prostate cancers, two breast cancers, one esophageal cancer, one brain cancer, one colorectal cancer, one kidney cancer, and forty-nine unspecified cancers (Appendix A). There were also five NMSCs identified in the exposed subjects not included in analysis. There were 57 malignancies in 26,154 patient-years of placebo and/or non-TNF-I drug exposure: thirty-two lymphomas, five prostate cancers, two lung cancers, and eighteen unspecified cancers (Appendix A). One NMSC was also identified in these subjects and not included.

The malignancy rate ratio for any malignancy was MRR 1.42 (95% CI 0.72–2.79) (Figure 4). For individual drugs, the rate ratios were as follows: ADA (MRR 1.19, 95% CI 0.44–3.22), INX (MRR 1.69, 95% CI 0.83–3.44), and ETN (MRR 1.85, 95% CI 0.50–6.89) (Figure 5A–C). There were no observational studies of certolizumab or golimumab meeting the inclusion criteria. Funnel plots did not demonstrate statistically significant asymmetry for overall TNF-I exposure or for any individual TNF-I exposure (Appendix A).

In the observational studies, 0/10 (0%) had adequate quality of adverse event reporting according to the Ioannidis and Lau criteria, and 8/10 (80%) had either industry or unknown funding sources (Table 2). In total, 2/10 (20%) of the observational studies were unlikely to have bias due to confounding, 10/10 (100%) were unlikely to have bias due to participant selection, 6/10 (60%) were unlikely to have bias due to the classification of interventions or deviation from intended interventions per the ROBINS-I criteria (Appendix A).

## 4. Discussion

We conducted a systematic review and meta-analyses of interventional and observational studies to evaluate the relationship between TNF-I exposure and the development of malignancies. This meta-analysis of 45 interventional studies did not demonstrate a statistically significant difference in cancer incidence for patients on a TNF-I (RR 1.04, 95% CI 0.71–1.51) (Figure 2). This meta-analysis of 10 observational studies also did not show a statistically increased risk of cancers in patients taking TNF-I (RR 1.42, 95% CI 0.72–2.79) (Figure 4) but did suggest a trend toward significance with both longer exposure to TNF-I and longer follow-up. This meta-analyses did not find differences in risk for patients taking specific TNF-I (Figure 3 and Figure 5), which is likely due to limited power when dividing the cohort up by individual drug exposure. Given its contemporary nature and scale, we believe the results presented here represent the broadest effort to date to define this relationship.

The most important finding of this study, however, is that the vast majority of the trials used to approve TNF-I indications, and the vast majority of long-term post-marketing studies, are inadequate in their quality of adverse event follow-up and have a significant risk for conflicts of interest and bias in reporting, with only 9 out of 55 studies (16.4%) demonstrating adequate adverse event reporting according to the Ioannidis and Lau criteria, only 3 out of 55 studies (5.5%) demonstrating clear non-industry funding, and only 40/55 (72.7%) demonstrating a low risk of bias per the Cochrane (interventional studies) and ROBINS-I (observational) criteria. Our meta-analysis is the first to report on these quality metrics for TNF-I studies and use the most stringent criteria for study inclusion. Many of the foundational efficacy randomized control trials were excluded due to the co-administration of drugs such as methotrexate or a trial design containing a drug crossover component. Another strength of our study is using patient-level data to collate and analyze individual and overall malignancy data.

The results of our meta-analyses are consistent with the antecedent literature and the findings of several prior meta-analyses, which did not demonstrate an effect of TNF-I use on the development of malignancies other than NMSC [15,17,30,31,32,33,34]. Several factors complicate the interpretation of these results. Most interventional studies of TNF-I were designed and powered to determine drug efficacy and rarely have follow-up greater than one year—as evidenced by the median length of follow-up of 41 weeks (IQR of 24–64 weeks) in our meta-analysis of the interventional studies. This may be an insufficient length of observation to define a difference in long-term cancer rates. The meta-analysis of observational studies incorporated datasets with a longer median follow-up of 82 weeks (IQR of 54–171 weeks) and indicated an increased risk of malignancies with TNF-I exposure, although not a statistically significant one. The meta-analyses of observational studies are challenging due to issues of publication bias and study and patient-level heterogeneity [27]. This was reflected in our quality of adverse events and risk of bias assessments (Table 2 and Appendix A) and could have potentially influenced our results despite attempts to restrict heterogeneity. Furthermore, few observational studies included in the meta-analysis directly compared a TNF-I to a placebo, with the comparator group usually being an anti-rheumatic medication other than a TNF-I (Table 1). Furthermore, there is no consensus in the literature as to what would be a biologically plausible timeframe post-drug exposure for malignancy development, which led to our decision to use 6 months post-exposure as a cutoff for biologic plausibility.

Our meta-analysis is also the first to our knowledge to employ this time-dependent follow-up analysis for drug exposure, which allowed us to meaningfully stratify patient cohorts into a model with significantly higher statistical power.

## 5. Conclusions

Our systematic review and meta-analyses of interventional and observational studies showed evidence of a relationship between TNF-I use and reported malignancies without reaching conventional thresholds for statistical significance. There were only a small number of high-quality studies with adequate adverse event reporting and an even smaller subset of those designed to investigate malignancy risk. These findings warrant further investigations over longer timeframes and the continued examination of the relationship between chronic inflammation, TNF-α signaling, and tumorigenesis.

## Figures and Tables

**Figure 1 cancers-17-00390-f001:**
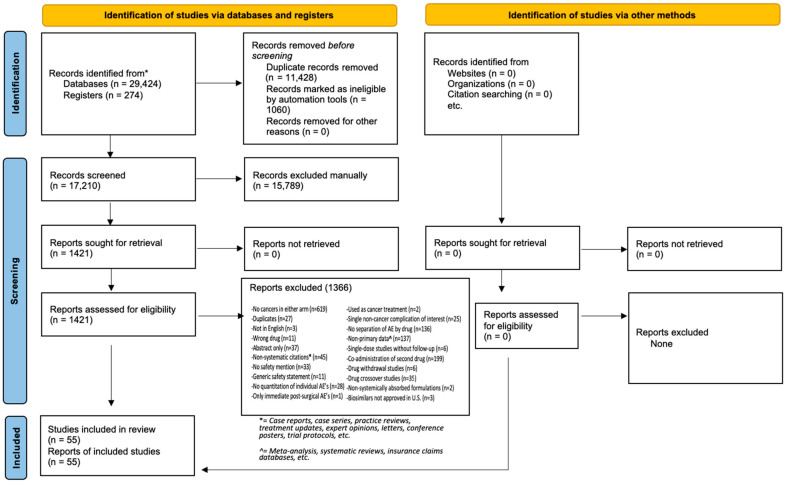
PRISMA flow diagram (2020).

**Figure 2 cancers-17-00390-f002:**
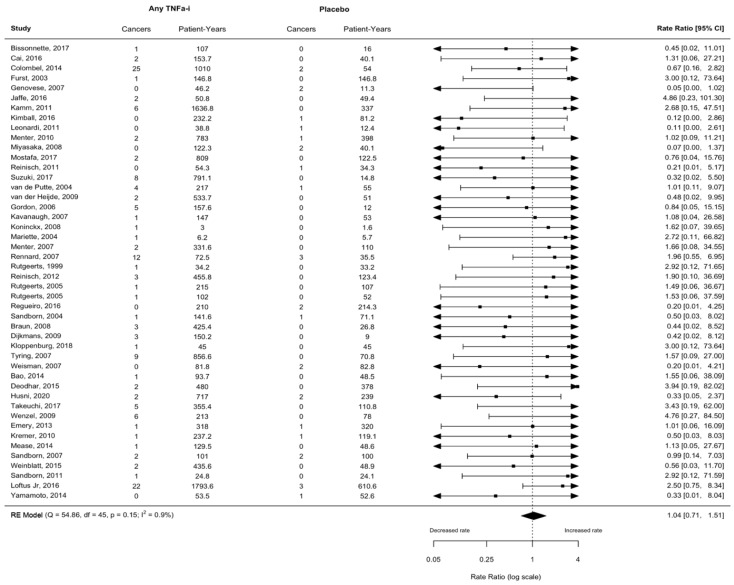
Malignancy rate ratio in patients exposed to any TNF-I excluding NMSC in interventional studies [47,48,49,50,51,52,53,54,55,56,57,58,59,60,61,62,63,64,65,66,67,68,69,70,71,72,73,74,75,76,77,78,79,80,81,82,83,84,85,86,87,88,89,90,91].

**Figure 3 cancers-17-00390-f003:**
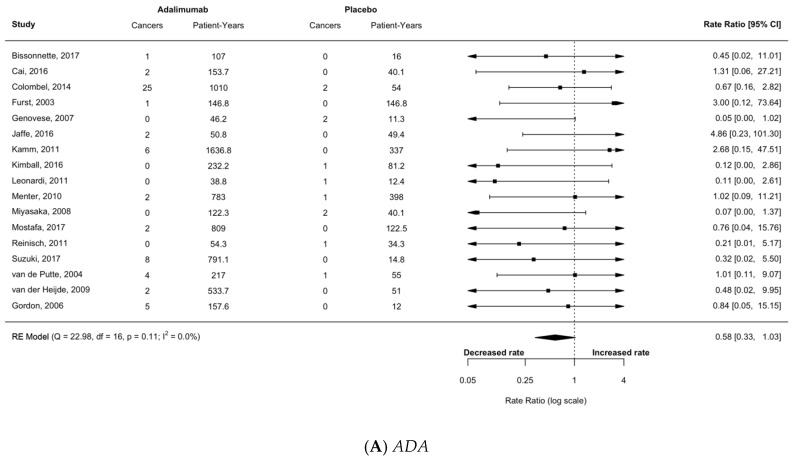
Calculated malignancy rate ratio in patients exposed to any TNF-alpha inhibitor (TNF-I) using cancers excluding non-melanoma skin cancer (NMSC) in 45 interventional studies for (**A**) adalimumab (ADA); (**B**) infliximab (INX); (**C**) etanercept (ETN); (**D**) certolizumab (CTZ); (**E**) golimumab (GOL) [47,48,49,50,51,52,53,54,55,56,57,58,59,60,61,62,63,64,65,66,67,68,69,70,71,72,73,74,75,76,77,78,79,80,81,82,83,84,85,86,87,88,89,90,91].

**Figure 4 cancers-17-00390-f004:**
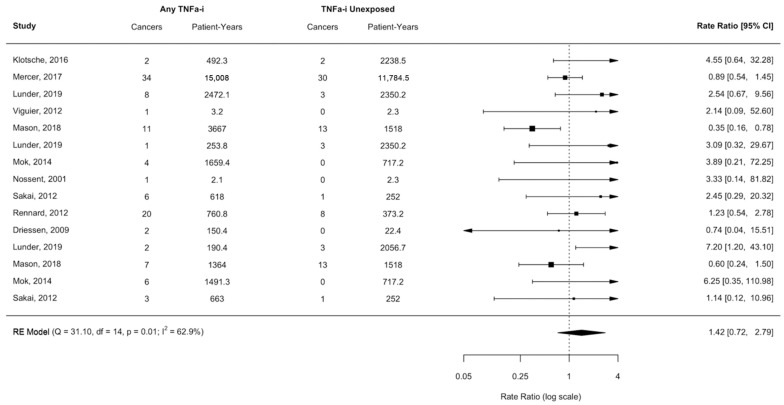
Malignancy rate ratio in patients exposed to any TNF-I excluding NMSC in observational studies [25,92,93,94,95,96,97,98,99,100].

**Figure 5 cancers-17-00390-f005:**
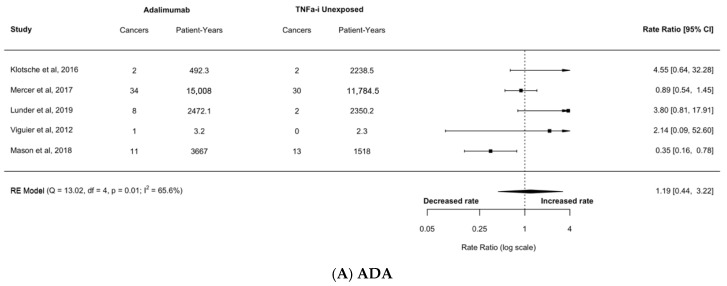
Calculated malignancy rate ratio in patients exposed to any TNF-alpha inhibitor (TNF-I) using cancers excluding non-melanoma skin cancer (NMSC) in 10 observational studies for (**A**) adalimumab (ADA); (**B**) infliximab (INX); (**C**) etanercept (ETN) [25,92,93,94,95,96,97,98,99,100].

**Table 1 cancers-17-00390-t001:** Interventional and observational study characteristics of patients exposed to TNF-I.

	Author, Year	Index Disease ^	Patients, Total No.	Patients, No. on TNF-a Inhibitor	Patients, No. Unexposed to TNF-a Inihibitor	Comparator Arm	Duration of Exposure, wk
	Interventional						
ADA ^							
	Bissonnette, 2017 [47]	PS	107	107	53	Placebo	52
	Cai, 2016 [48]	PS	425	333	87	Placebo	24
	Colombel, 2014 [49]	UC	1093	1010	54	Placebo	52
	Furst, 2003 [50]	RA	636	318	318	Placebo	24
	Genovese, 2007 [51]	PSA	100	100 *	49 *	Placebo	24
	Jaffe, 2016 [52]	Noninfectious uveitis	217	110	107	Placebo	24
	Kamm, 2011 [53]	CD	778	519	107	Placebo	164
	Kimball, 2016 [54]	HS	633	361	252	Placebo	36
	Leonardi, 2011 [55]	PS	72	72 *	23 *	Placebo	28
	Menter, 2010 [56]	PS	1212	783	398	Placebo	52
	Miyasaka, 2008 [57]	RA	352	265 *	87 *	Placebo	24
	Mostafa, 2017 [58]	PS	1212	809	398	Placebo	52
	Reinisch, 2011 [59]	UC	576	353	223	Placebo	8
	Suzuki, 2017 [60]	UC	274	266 *	96 *	Placebo	196
	van de Putte, 2004 [61]	RA	544	434 *	110 *	Placebo	26
	van der Heijde, 2009 [62]	AS	315	311	107	Placebo	104
	Gordon, 2006 [63]	PS	147	147	52	Placebo	60
INX ^							
	Kavanaugh, 2007 [64]	PSA	200	100	100	Placebo	52
	Koninckx, 2008 [65]	Endometriosis	21	13	7	Placebo	12
	Mariette, 2004 [66]	SS	103	54	49	Placebo	6
	Menter, 2007 [67]	PS	835	627	208	Placebo	27.5
	Rennard, 2007 [68]	COPD	234	157 *	77 *	Placebo	24
	Rutgeerts, 1999 [69]	CD	73	37	36	Placebo	48
	Reinisch, 2012 [70]	UC	291	229 *	62 *	Placebo	103.5
	Rutgeerts, 2005 [71]	UC	364	243 *	121	Placebo	46
	Rutgeerts, 2005 [71]	UC	364	241 *	123	Placebo	22
	Regueiro, 2016 [72]	CD	297	147	150	Placebo	74.3
	Sandborn, 2004 [73]	CD	396	263	132	Placebo	28
	Braun, 2008 [74]	AS	279	201	78	Placebo	102
ETN ^							
	Dijkmans, 2009 [75]	AS	84	81 *	39 *	Placebo	108
	Kloppenburg, 2018 [76]	OA	90	45	45	Placebo	52
	Tyring, 2007 [77]	PS	591	464 *	307 *	Placebo	96
	Weisman, 2007 [78]	RA	535	266	269	Placebo	16
GLM ^							
	Bao, 2014 [79]	AS	213	203	105	Placebo	24
	Deodhar, 2015 [80]	AS	356	258	78	Placebo	252
	Husni, 2020 [81]	PSA	480	480 *	239 *	Placebo	52
	Takeuchi, 2017 [82]	RA	316	154 *	105 *	Placebo	120
	Wenzel, 2009 [83]	Asthma	309	231 *	78 *	Placebo	52
	Emery, 2013 [84]	RA	637	159	160	Placebo + MTX	104
	Kremer, 2010 [85]	RA	643	257	129	Placebo + MTX	48
CTZ ^							
	Mease, 2014 [86]	PSA	409	159	136	Placebo	24
	Sandborn, 2007 [87]	CD	662	202	329	Placebo	26
	Weinblatt, 2015 [88]	RA	1063	809 *	212 *	Placebo	28
	Sandborn, 2011 [89]	CD	439	215	209	Placebo	6
	Loftus Jr, 2016 [90]	CD	3445	2570	875	Placebo	36.29
	Yamamoto, 2014 [91]	RA	230	116	114	Placebo	24
	Observational						
ADA ^							
	Klotsche, 2016 [92]	JIA	3189	320	1455	Methotrexate, ETN	80
	Mercer, 2017 [25]	RA	15,298	4288	3367	DMARD, ETA, IFX	182
	Lunder, 2019 [93]	PS	1606	750	713	Ixekizumab, Apremilast, Secukinumab, Ustekinumab	171.4
	Viguier, 2012 [94]	PS	28	7	5	Ustekinumab, Efalizumab, IFX, ETN	24
	Mason, 2018 [95]	PS	8533	3667	1518	Ustekinumab	52
INX ^							
	Lunder, 2019 [93]	PS	1606	77	713	Ixekizumab, Apremilast, Secukinumab, Ustekinumab	171.4
	Mok, 2014 [96]	RA	1769	671	290	Abatacept, Belimumab, Rituximab, Tocilizumab	128.6
	Nossent, 2001 [97]	Polyarthritis	19	9	10	Methylprednisolone	12
	Sakai, 2012 [98]	RA	1028	412	168	Tocilizumab	78
	Rennard, 2012 [99]	COPD	234	157	77	Placebo	252
ETN ^							
	Driessen, 2009 [100]	PS	118	94	14	Efalizumab	83.2
	Lunder, 2019 [93]	PS	1606	66	713	Apremilast, Ixekizumab, Secukinumab, Ustekinumab	171.4
	Mason, 2018 [95]	PS	8533	1364	1518	Ustekinumab	52
	Mok, 2014 [96]	RA	1769	603	290	Abatacept, Belimumab, Rituximab, Tocilizumab	128.6
	Sakai, 2012 [98]	RA	1028	442	168	Tocilizumab	78

* manually calculated due to initial time spent in placebo arm. ^ adalimumab (ADA), infliximab (INX), etanercept (ETN), certolizumab (CTZ), golimumab (GLM), psoriasis (PS), psoriatic arthritis (PSA), ulcerative colitis (US), Crohn’s disease (CD), hidradenitis suppurativa (HS), rheumatoid arthritis (RA), Sjogren’s syndrome (SS), chronic obstructive pulmonary disease (COPD), ankylosing spondylitis (AS), osteoarthritis (OA), juvenile idiopathic arthritis (JIA).

**Table 2 cancers-17-00390-t002:** Quality of adverse event reporting (Ioannidis and Lau criteria) [33]: Interventional studies and observational studies.

		Quality of Adverse Event Reporting	Adequacy of Blinding	Conflicts of Interest	Funding Source
ADA *					
	Bissonnette, 2017 [47]	Partially adequate	Yes	Yes	Industry
	Cai, 2016 [48]	Partially adequate	Yes	No	Industry
	Colombel, 2014 [49]	Partially adequate	Not Applicable	Unknown	Unknown
	Furst, 2003 [50]	Adequate	Yes	Yes	Industry (Abbott)
	Genovese, 2007 [51]	Partially adequate	Unknown	Unknown	Unknown
	Jaffe, 2016 [52]	Partially adequate	Unknown	Unknown	Industry
	Kamm, 2011 [53]	Partially adequate	Yes	Yes	Industry
	Kimball, 2016 [54]	Adequate	Yes	Unknown	Unknown
	Leonardi, 2011 [55]	Inadequate	Yes	Unknown	Industry
	Menter, 2010 [56]	Partially adequate	Not Applicable	Unknown	Industry
	Miyasaka, 2008 [57]	Inadequate	Yes	Yes	Industry (Abbott)
	Mostafa, 2017 [58]	Partially adequate	Yes	Unknown	Industry
	Reinisch, 2011 [59]	Partially adequate	Yes	Unknown	Industry
	Suzuki, 2017 [60]	Partially adequate	Not Applicable	Unknown	Industry
	van de Putte, 2004 [61]	Partially adequate	Yes	Yes	Industry (Abbott)
	van der Heijde, 2009 [62]	Adequate	Yes	Unknown	Industry
	Gordon, 2006 [63]	Adequate	Not Applicable	Unknown	Industry
INX *					
	Kavanaugh, 2007 [64]	Inadequate	Yes	Yes	Industry
	Koninckx, 2008 [65]	Inadequate	Yes	Unknown	University grant
	Mariette, 2004 [66]	Inadequate	Yes	Unknown	Unknown
	Menter, 2007 [67]	Adequate	Yes	Unknown	Unknown
	Rennard, 2007 [68]	Partially adequate	Not Applicable	Unknown	Industry
	Rutgeerts, 1999 [69]	Partially adequate	Yes	Unknown	Unknown
	Reinisch, 2012 [70]	Partially adequate	Not Applicable	Unknown	Unknown
	Rutgeerts, 2005 [71]	Adequate	Yes	Yes	Industry
	Rutgeerts, 2005 [71]	Adequate	Yes	Yes	Industry
	Regueiro, 2016 [72]	Partially adequate	Not Applicable	Yes	Industry
	Sandborn, 2004 [73]	Adequate	Yes	Yes	Industry
	Braun, 2008 [74]	Adequate	Yes	Yes	Industry
ETN *					
	Dijkmans, 2009 [75]	Partially adequate	Yes	Unknown	Industry
	Kloppenburg, 2018 [76]	Partially adequate	Yes	Unknown	Industry
	Tyring, 2007 [77]	Partially adequate	Yes	Yes	Industry
	Weisman, 2007 [78]	Partially adequate	Yes	Yes	Industry (Amgen)
GOL *					
	Bao, 2014 [79]	Partially adequate	Yes	No	Industry
	Deodhar, 2015 [80]	Partially adequate	Yes	Unknown	Industry
	Husni, 2020 [81]	Partially adequate	Yes	Unknown	Unknown
	Takeuchi, 2017 [82]	Partially adequate	Yes	Unknown	Unknown
	Wenzel, 2009 [83]	Partially adequate	Yes	Unknown	Industry
	Emery, 2013 [84]	Partially adequate	Not Applicable	Unknown	Unknown
	Kremer, 2010 [85]	Partially adequate	Yes	Unknown	Industry
CTZ *					
	Mease, 2014 [86]	Partially adequate	Yes	Unknown	Unknown
	Sandborn, 2007 [87]	Partially adequate	Yes	Unknown	Unknown
	Weinblatt, 2015 [88]	Partially adequate	Yes	Unknown	Industry
	Sandborn, 2011 [89]	Adequate	Yes	Unknown	Industry
	Loftus Jr, 2016 [90]	Partially adequate	Yes	Unknown	Industry
	Yamamoto, 2014 [91]	Partially adequate	Yes	Unknown	Industry
ADA ^					
	Klotsche, 2016 [92]	Partially adequate	Not Applicable	Unknown	Industry
	Mercer, 2017 [25]	Partially adequate	Not Applicable	Unknown	Government grant
	Lunder, 2019 [93]	Partially adequate	Not Applicable	Unknown	Industry
	Viguier, 2012 [94]	Inadequate	Not Applicable	Yes	Unknown
	Mason, 2018 [95]	Partially adequate	Unknown	Unknown	Unknown
INX ^					
	Lunder, 2019 [93]	Partially adequate	Not Applicable	Unknown	Industry
	Mok, 2014 [96]	Partially adequate	Not Applicable	Unknown	Unknown
	Nossent, 2001 [97]	Partially adequate	Not Applicable	Unknown	Unknown
	Sakai, 2012 [98]	Partially adequate	Not Applicable	Unknown	Government grant
	Rennard, 2012 [99]	Partially adequate	Not Applicable	Unknown	Industry
ETN ^					
	Driessen, 2009 [100]	Partially adequate	Not Applicable	Yes	Industry
	Lunder, 2019 [93]	Partially adequate	Not Applicable	Unknown	Industry
	Mason, 2018 [95]	Partially adequate	Unknown	Unknown	Unknown
	Mok, 2014 [96]	Partially adequate	Not Applicable	Unknown	Unknown
	Sakai, 2012 [98]	Partially adequate	Not Applicable	Unknown	Government grant

* Interventional study; ^ Observational study.

## Data Availability

The dataset supporting the conclusions of this article is available in the GitHub repository, https://github.com/conordriscoll/TNF-Alpha-Meta-Analysis.git (accessed on 15 July 2020).

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
