# Peer review of "Tumor Necrosis Factor-Alpha Inhibitor Use and Malignancy Risk: A Systematic Review and Patient Level Meta-Analysis"

_cancers, 2025, doi:10.3390/cancers17030390_

Round 1
Reviewer 1 Report
Comments and Suggestions for Authors
Dear authors, I read with interest your manuscript. Despite no statistical significant correlation between the TNF-I use and the development of secondary malignancies, I think your article deserves attention. I found no major or minor issues regarding the methodology or data presentation, so my recommendation is to accept the paper in its present form.
Best regards
Author Response
Comments 1: Dear authors, I read with interest your manuscript. Despite no statistical significant correlation between the TNF-I use and the development of secondary malignancies, I think your article deserves attention. I found no major or minor issues regarding the methodology or data presentation, so my recommendation is to accept the paper in its present form.
Response 1: No changes needed. Thank you for the kind feedback.
Reviewer 2 Report
Comments and Suggestions for Authors
This is an interesting paper, but is limited by the small retrospective and observational studies and the usual limitations of systematic reviews and meta-analyses. Difficult to draw conclusions. The objectives and methods are clearly presented, and data is well described. Overall, it is easy to read, design and statistics are appropriate, discussion and conclusions reflect the study
Author Response
Comments 1: This is an interesting paper, but is limited by the small retrospective and observational studies and the usual limitations of systematic reviews and meta-analyses. Difficult to draw conclusions. The objectives and methods are clearly presented, and data is well described. Overall, it is easy to read, design and statistics are appropriate, discussion and conclusions reflect the study
Response 1: No edits requested. Thank you for your review.
Reviewer 3 Report
Comments and Suggestions for Authors
The authors´ performed a comprehensive evaluation (systematic review/patient-level metanalysis) on the malignancy risk (against 146 malignancies) in patients exposed (15,120 patient-years of exposure) to TNF-α inhibitors [TNF-I; particularly Adalimumab (ADA), a monoclonal antibody with dual action]. After a very well-orchestrated and detailed data screening (MeSH unique IDs, PRISMA 2000; Sections 2.1-2,4) and statistical evaluation (supplementary material, section 2.5) of 45 interventional and 10 observational studies (34, 866 patients) published between 1996-2020. The authors’ analysis indicate that TNF-I exposure was not associated with overall 59 cancer risk (other than non-melanoma skin cancer) in both interventional and observational (RR= 1.04-1.42), backing up the safe use of TNF-I for controlling inflammatory-related diseases. Though the manuscript`s quality is good enough to be considered for publication, minor changes may improve even more its uniqueness and scientific soundness:
· General. A) Include the meaning of each abbreviation the first time it is mentioned and, if possible, reduce their use throughout the manuscript (e.g. FAERS). B) Suggestion: Use MRR (or a more appropriate abbreviation) for malignancy rate ratio to avoid misunderstandings with risk ratio (also RR).
· Title. OK
· Simple summary. Please eliminate non-evidence-based statements [a possible conflicting result with lymphoma, (line 36) not evidenced in abstract]. Do the same throughout the manuscript.
· Abstract/Introduction. OK.
· Methods. A) Section 2.1 - Include MeSH unique ID`s for each term used. B) Section 2.2 - Authors should give a very brief explanation as to why they restricted the search only to US approved drugs (were all studies conducted in US as well?).
· Results. A) Please include each figure or table immediately after its corresponding mention within the text and, consequently, eliminate section 3.4. B) Sections 3.2-3.3- Considering that TNF-I dosage was recorded (section 2.3), mean dosages (and protocol of administration) for each drug should me mentioned
· Tables & figures. A) All images and tables should be formatted according to Cancers´ guidelines. B) All figures should be provided at enough size and resolution (>300 dpi) C) The titles and footnotes must be detailed enough to allow their agile and independent reading from the text. Do the same for supplementary materials. C) Tables 1 and 2 should be submitted as supplementary materials and bring back to the main text any important supplementary information (e.g. ADA-related data).
· Supplementary materials. A) Use codes for rating Tables S1 and S2, otherwise align them horizontally.
· Discussion. A) 10-15 lines long yet "effective" paragraphs are suggested (as you did in introduction section). B) Seems to short and not comparative to preceding studies (References mentioned in introduction section, lines 89-113.
· Conclusions. OK
· References. A) Several references do not have the appropriate format for this journal. B)
Author Response
Comments 1: General. A) Include the meaning of each abbreviation the first time it is mentioned and, if possible, reduce their use throughout the manuscript (e.g. FAERS). B) Suggestion: Use MRR (or a more appropriate abbreviation) for malignancy rate ratio to avoid misunderstandings with risk ratio (also RR).
Response 1: Low usage abbreviations were removed and all instances where RR was written were converted to MRR.
Comments 2: Simple summary. Please eliminate non-evidence-based statements [a possible conflicting result with lymphoma, (line 36) not evidenced in abstract]. Do the same throughout the manuscript.
Response 2: The line about lymphoma was removed in the abstract and non-evidence-based statements were removed.
Comments 3: Methods. A) Section 2.1 - Include MeSH unique ID`s for each term used. B) Section 2.2 - Authors should give a very brief explanation as to why they restricted the search only to US approved drugs (were all studies conducted in US as well?).
Response 3: (A) MeSH Unique ID’s are now included for each term used in the Supplementary material. (B) The same five drugs are approved in Europe and Asia so it now reads “approved for use in the United States, Europe, or Asia” as that covers all of the associated trials.
Comments 4: Results. A) Please include each figure or table immediately after its corresponding mention within the text and, consequently, eliminate section 3.4. B) Sections 3.2-3.3- Considering that TNF-I dosage was recorded (section 2.3), mean dosages (and protocol of administration) for each drug should me mentioned
Response 4: (A) All Figures and Tables have been to the section where they are first mentioned as requested, section 3.4 has been eliminated. (B) median duration of exposure has been added to both Interventional studies in Section 3.2 and Observational studies in Section 3.3. We added the route of administration to the Methods section 2.2 Inclusion and Exclusion criteria, writing “The two medications administered intra-venously are INX (3mg/kg every 8 weeks) and GLM (100mg every 4 weeks after induction) are administered intra-venously. The three medications administered subcutaneously are ADA (40mg every other week), ETN (total of 50mg weekly), and CTZ (total of 400mg every 4 weeks after induction).” as these were their only tested formulations.
Comments 5: Tables & figures. A) All images and tables should be formatted according to Cancers´ guidelines. B) All figures should be provided at enough size and resolution (>300 dpi) C) The titles and footnotes must be detailed enough to allow their agile and independent reading from the text. Do the same for supplementary materials. C) Tables 1 and 2 should be submitted as supplementary materials and bring back to the main text any important supplementary information (e.g. ADA-related data).
Response 5: (A/B/C) All formatting is now correct for the Cancers journal. We exchanged tables and figures (some from main text to supplementary files and others the opposite) per specifications to allow for smoother understanding of the main text.
Comments 6: Supplementary materials. A) Use codes for rating Tables S1 and S2, otherwise align them horizontally.
Response 6: Both Tables S1 and S2 have been aligned horizontally.
Comments 7: Discussion. A) 10-15 lines long yet "effective" paragraphs are suggested (as you did in introduction section). B) Seems to short and not comparative to preceding studies (References mentioned in introduction section, lines 89-113.
Response 7: (A) this has been reworked to the structure recommended here. (B) Discussion section has be re-worked to emphasize the finding of most studies having inadequate adverse event reporting, which is then placed in the context of the preceding studies
Comments 8: References. A) Several references do not have the appropriate format for this journal. B)
Response 8: All references have been edited meticulously to adhere to journal format.
Reviewer 4 Report
Comments and Suggestions for Authors
The onset of cancer during TNFin treatment is a very important topic. Your review has a good structure and excellent statistical analysis. It seems a really comprehensive review. Also Tables and Figures are useful to read the paper.
Maybe you should add in the intro section if literature suggest to withdraw therapy with TNFin in caso of new onset of cancer. A recent review paper about Denaro N et al, published on Int J Mol Sci takes into account also this topic in psoriatic patients.
Author Response
Comments 1: Maybe you should add in the intro section if literature suggest to withdraw therapy with TNFin in caso of new onset of cancer. A recent review paper about Denaro N et al, published on Int J Mol Sci takes into account also this topic in psoriatic patients.
Response 1: This comment has been addressed in the Introduction with the recommended paper cited as the new reference #30.
Reviewer 5 Report
Comments and Suggestions for Authors
The review article analyzed the association of TNF-I with cancer and concluded that use of TNF-I is not significantly associated with risk of carcinoma. The article is well written, and the conclusion is supported by the data included. The incidence of malignancy reported in previous studies have been pooled and analyzed, was there any difference based on gender. The authors mention in the limitation- short follow up. TNF-I may generally be more effective in male patients compared to female patients, with women often experiencing a higher rate of side effects and potentially needing to discontinue treatment earlier due to lack of efficacy or increased adverse events like infections. Discontinuation of TNF-I was considered or not- this may be a confounding factor. Differential effect may be due to testosterone and estrogen, it is not clear from the data whether females/males were on any hormonal therapy? status of the disease when TNF-I was used may also affect outcome- whether this was considered in the original studies? These concerns need to be described in the review article or in the exclusion/inclusion section or in limitations.
Author Response
Comments 1: TNF-I may generally be more effective in male patients compared to female patients, with women often experiencing a higher rate of side effects and potentially needing to discontinue treatment earlier due to lack of efficacy or increased adverse events like infections. Discontinuation of TNF-I was considered or not- this may be a confounding factor. Differential effect may be due to testosterone and estrogen, it is not clear from the data whether females/males were on any hormonal therapy? status of the disease when TNF-I was used may also affect outcome- whether this was considered in the original studies? These concerns need to be described in the review article or in the exclusion/inclusion section or in limitations.
Response 1: All of the points from this comment were included in the Discussion section as limitations.